# Grateful Client Philanthropy and Veterinary Medicine: Experiences at North Carolina State University

**DOI:** 10.3390/vetsci6020044

**Published:** 2019-05-16

**Authors:** Dianne Dunning, Sherry L. Buckles, David C. Dorman

**Affiliations:** North Carolina Veterinary Medical Foundation, College of Veterinary Medicine, North Carolina State University, Raleigh, NC 27607, USA; slbuckle@ncsu.edu (S.L.B.); david_dorman@ncsu.edu (D.C.D.)

**Keywords:** grateful patient, grateful client, philanthropy, fundraising, veterinary medicine, Academia

## Abstract

The historical reliance of state and federal funds as a sole source of veterinary educational activities has created a funding gap at many academic institutions. Due to declining resources, philanthropy has become an important source of financial support for veterinary colleges in the United States. In particular, for academic institutions with veterinary hospitals, grateful client philanthropy has been an increasingly important area of resource growth. Philanthropic gifts support innovative research, scholarship and capital, and programmatic initiatives. Areas of giving are often geared towards major infrastructure gifts and naming opportunities, faculty endowment, student scholarships, and other gift opportunities. This review provides an overview of grateful client philanthropy at North Carolina State University College of Veterinary Medicine and explores the various giving opportunities and challenges of donor giving in veterinary medicine. (129/200)

## 1. Introduction

Many veterinary colleges in the United States (U.S.) are located at land-grant universities and were established to address issues related to production agriculture. Veterinary schools provide several critical societal functions including education of the next generation of veterinarians, they contribute basic and applied research, and they offer care to veterinary patients and their owners. Historically, veterinary educational activities at many of these veterinary colleges were initially fully funded by a combination of state and federal funds [1]. Today, financial support for many veterinary colleges is derived from several sources including state or institutional resources, clinical and diagnostic activities, research grants and contracts, and philanthropy. The economic downturn of the early 2000′s created a funding gap at many academic institutions. This recession led many U.S. states and the federal government to sharply reduce funding of public universities and their veterinary colleges [1,2,3]. When adjusted for inflation, current (2018) research funding levels provided by the U.S. National Institutes of Health (NIH) are approximately $2 billion lower than what was available in the 2003 federal budget [4]. Declining state and federal support for faculty positions, research, and tuition support have led to reduced numbers of research grants, reduced hiring, layoffs, and program elimination at U.S. veterinary colleges [2]. Declining support has also contributed to the rising cost of veterinary education.

These economic pressures have prompted academic institutions to increasingly turn to philanthropic giving as a revenue stream. For example, the Association of American Medical Colleges (AAMC) reports that endowments and gifts to the 139 fully-accredited U.S. medical schools were approximately $4.8 billion in 2016–2017 [5]. These gifts represented 3.7% of all revenues collected by these 139 medical schools in this timeframe [5]. The income from endowments and gifts was similar to revenues derived from tuition and fees at the surveyed medical schools [6]. Similarly, the Association for Healthcare Philanthropy (AHP) reports that in 2017, non-profit hospitals and health care organizations raised over $10.4 billion dollars through philanthropic giving [7]. Investments in development have a favorable return on investment. For example, the AAMC reports that fundraising efforts by U.S. medical schools had an approximate 8 to 1 return on investment in 2016-2017 [8,9]. According to the AHP, fundraising efforts by non-profit hospitals and health care organizations had an approximate 4 to 1 return on investment [7].

One form of philanthropy in human medicine, grateful patient philanthropy, has been described as donations that are made in recognition of excellent medical care received [4]. In human medicine, grateful patient philanthropy is an important source of philanthropic giving [10] and a number of institutional best practices have been promulgated [11,12,13]. The AAMC’s 2017 Annual Development Survey reports that individuals who are not alumni or staff of medical schools or teaching hospitals remained the largest segment of individual donors (84%) with many of these gifts being likely made by patients and/or family members of patients [9]. The goal of these fund raising efforts is to turn the positive feelings of grateful patients into support for new research, endowed professorships, academic scholarships, capital improvements, and support for other institutional and programmatic needs [11,12,13].

The subject of philanthropy in veterinary education has largely been ignored in the veterinary medical literature. This article describes current philanthropic efforts that have evolved at the College of Veterinary Medicine (CVM) at North Carolina State University (NC State). Successes as well as problems encountered will be discussed and research needs identified. For convenience, the manuscript is organized into different gift types (e.g., Non-major gifts and major, leadership and principal gifts). Gift ranges and capacity ratings enable development officers to strategize and match a donor’s priorities and giving potential with institutional needs. Capacity ratings are particularly important in developing a donor gift plan and engagement/solicitation strategies, as they represent an informed estimate of a donor’s ability to give. Donor gift capacity is usually based on public financial information and utilizes data on traditional wealth indicators such as previous philanthropic giving, business affiliations, real estate and other assets. Donor gift capacity are by their nature, an incomplete picture of an individual’s true gift capacity. Therefore, they should never be used in isolation as they do not offer any insight as to the donor experience or their desire to make an impact or recognize an individual or program. As a result, they will never offer an exact picture of an individual’s priorities or philanthropic propensity to a specific organization. These drivers can only be identified through ongoing discussion with prospects. In this article, where appropriate, we will discuss similar efforts at other U.S. colleges of veterinary medicine. For the purposes of this discussion, we treat grateful client philanthropy as the veterinary equivalent of grateful patient philanthropy in human medicine.

## 2. Non-major Gifts in Grateful Client Philanthropy

For the purposes of this review, non-major gifts are most frequently encountered and are further defined as gifts with a cash value of less than $50,000. Donors to an academic institution may designate or “restrict” the use of their donations to a particular purpose or project. In contrast, “unrestricted” gifts are donations that the institution may use for any purpose.

### 2.1. Unrestricted Gifts (examples at NC State CVM Include Our All Gifts Great and Small, Walk of Honor, Gallop of Honor, and Pets in Memorium Programs)

Any robust development program should include an annual giving mechanism for donors to give smaller amounts to a veterinary school. Gift recognition programs, such as a memorial pet program, often provide the donor with an initial easy and satisfying way to recognize their pet, family member, clinician/faculty member, or student. The NC State CVM developed a Walk of Honor program in 2003 that allows donors to designate a brick or a paver in the main pedestrian pathways of the CVM campus. Designating a brick or 18” square stone paver currently requires a gift of $200 or $500 or more, respectively. A companion program, the Gallop of Honor requires a gift of $250 or more for a personalized horseshoe that is displayed in the CVM’s equine medicine facilities. Over $830,000 in net revenue has been raised since the inception of these specific programs. Collectively, annual donations to the NC State CVM’s general unrestricted fund in the fiscal year 2018 totaled $300,000, which includes the All Gifts Great and Small Fund. This pipeline of giving also helps identify donors and grateful clients who may not show up on a capacity rating report, but who have, through their unrestricted gift, indicated they are interested in further engagement.

### 2.2. Restricted Gifts in Recognition of Faculty, Staff, Students and Clinical Service (Examples at NC State CVM Include Our White Coat Program or Hospital Equipment)

After unrestricted giving, the most common form of grateful client giving, is a restricted gift in recognition of faculty, staff, students and clinical service. Many clients express their gratitude for the compassionate care they received for their animal in the form of a naming gift, programmatic support, or purchase of equipment. Therefore, NC State CVM has created additional programs that allow donors to recognize employees and students at the veterinary school. For example, in 2005 the college developed the “Coat of Excellence” program that allows donors to name a white laboratory coat in honor of an individual at the CVM. A recent naming opportunity involved Dr. Lysa Posner, a veterinary anesthesiologist who was instrumental in saving the life of a dog during a routine CT imaging session. To date, over 35 Coats of Excellence have been awarded to NC State faculty and staff. Currently, the Coat of Excellence program requires a minimum gift of $10,000 or more payable as either as an outright gift or pledged over two years. These gifts are allocated in the following way: 50% to the honoree’s service area; 25% to the NC State Veterinary Hospital to support client services, staff incentives, and areas of need; 15% to the honoree as a cash award; and 10% to the North Carolina Veterinary Medical Foundation (NCVMF) to support the operational costs of the program.

Another form of restricted giving is our clinical equipment list, which is generated and prioritized by the NC State Veterinary Hospital Board. In total, this list currently identifies approximately $1.7 million in hospital equipment needs. Individual equipment items often cost between $10,000 and $40,000 to purchase and are associated with individual clinical services and need (i.e., immediate need, priority need but not urgent and can be delayed until next financial year). At NC State CVM, we have a cadre of donors that request our hospital equipment list on an annual basis and use it as a guide to their giving based upon their ongoing experiences with hospital services. After the gift is made and the equipment is purchased, the donor has the option of visiting and seeing the equipment in use and its impact on improved patient care.

Other donors choose to support the faculty, staff and students in a clinical service by giving to professional development training and continuing education programs. One of NC State’s CVM longest standing donor-supported continued education program is the Shasta Rhodes Cardiology Seminar on Cardiovascular Health. This annual seminar is supported by grateful cardiology clients and is geared to resident education. Every year, the college’s cardiology faculty invites a respected guest lecturer for an innovative seminar and breakfast on the latest science in cardiology. This event is valued by the donor and the entire service and has led to additional support for the service and their priority needs.

## 3. Major, Leadership and Principal Gifts in Grateful Client Philanthropy

At NC State, a major gift is defined as a gift over $50,000. This is also the required level for creating an endowment to support an individual or program in perpetuity. A leadership and principal gift is defined as $1,000,000 to $4,999,999, and greater than $5,000,000, respectively.

### 3.1. Student Scholarship

One of the most common major gift opportunities is an endowed student scholarship. Student scholarships are critical to the financial well-being of many individual students. Scholarships help reduce the burden of debt of veterinary students and helps the institution maintain and grow its reputation as a global leader in shaping the future of veterinary medicine for the benefit of animal and human health. Scholarship support is increasingly important since the average educational debt over four years for 2016 graduates of veterinary school was $138,000, including those that graduated with no debt [14]. In 2015, the NC State CVM received $8 million from the R.B. Terry Charitable Foundation to support student scholarships. This gift had a tremendous impact as it nearly doubled the size of the endowments at the NC State CVM dedicated for scholarships.

### 3.2. Faculty Endowment, Programmatic and Capital Gifts

Another major gift opportunity is endowments that support individual faculty or groups of faculty. Endowed positions enable universities to recruit and retain the best and brightest faculty. Some endowments provide salary support while others provide discretionary funds to support faculty research, travel, and other professional needs. Endowed positions also provide an opportunity to recognize the value and importance of a particular faculty member. It has also been our experience that donors who have committed major gifts to support capital projects have also supported endowed chairs. For example, a 2017 gift from the R.B. Terry Charitable Foundation included $5 million to endow professorships.

At NC State, there are two levels of faculty endowment that exist: an endowed chair and an endowed distinguished professorship. These require single gifts of $2.5 million and $1 million, respectively. At NC State CVM, there were no fully endowed faculty in 2009, while today, the College has eight endowed professorships and chairs. Endowed positions within the NC State CVM are often aligned with the interests of the donor. For example, the Dr. Kady M. Gjessing and Rahna M. Davidson Distinguished Chair in Gerontology which is held by Dr. Natasha Olby, a NC State CVM faculty member, reflects the family’s interest in older animals. A second NC State CVM faculty member, Dr. Jody Gookin, is the holder of the title of the FluoroScience Distinguished Professor in Veterinary Scholars Research Education. This endowment supports activities related to the college’s summer research internship program.

Another major gift opportunity is capital gifts whose required minimum is $5 million for new buildings, facility renovations, or similar capital improvements. These gifts are often transformative and can lead to rapid improvement in teaching, research, and clinical capabilities. NC State CVM was the beneficiary of an initial $20 million gift from the R.B. Terry Charitable Foundation [15]. A portion of this gift was used to help build the Randall B. Terry, Jr. Companion Animal Veterinary Medical Center. Completion of the $72 million Terry Center included additional appropriations from the State of North Carolina [15]. A co-publisher of the High Point Enterprise, Mr. Terry first became involved with the NC State CVM in 1998 when one of his nine beloved golden retrievers, Nike, fell ill [16]. The superb care Nike received at NC State prompted Terry to join and later preside over the NCVMF. In the final years of his life, he was devoted to the betterment of veterinary medicine [16]. Since Mr. Terry’s death in 2005, the R.B. Terry Charitable Foundation has given over $51 million dollars to NC State CVM [16,17]. Most recently, the R.B. Terry Charitable Foundation has completed a $16 million dollar pledge, which supports endowed scholarships, professorships and research needs in addition to unrestricted giving toward vital infrastructure needs [17]. This lasting legacy of giving perhaps best exemplifies the power of grateful client giving and its real and tangible value and transformational benefits.

## 4. General Discussion

Developing a modern development program requires relationship building between motivated donors, dedicated fundraising staff, administrators, faculty and staff [18]. Donors are increasingly sophisticated and often see their gifts from a programmatic perspective [19]. In addition, modern donors are increasingly technologically savvy and simple online fundraising tools need to be developed to encourage giving [20].

Experiences at NC State CVM are common to other academic veterinary colleges. Many U.S. veterinary colleges have development officers and senior administrators (e.g., Assistant or Associate Deans) that oversee philanthropic activities. All thirty of the American Veterinary Medical Association (AVMA) accredited veterinary colleges located in the U.S. have a direct link on their internet home page that prompts a website visitor to give to the college. Donors visiting websites for U.S. veterinary colleges can choose from a range of giving opportunities that are qualitatively similar to those that exist at NC State CVM. In contrast, websites for European AVMA accredited veterinary colleges do not uniformly include links for giving opportunities on their home pages. In addition, U.S. veterinary colleges have also been the recipients of major donor gifts. For example, recent examples of principal gifts at U.S. veterinary schools include a $9 million donation by the Stanton Foundation to The Ohio State University CVM to construct a new Clinical and Professional Skills Lab [21]. This laboratory provides simulation models, instruction space for 80 students, and seven mock examination rooms. Cora Nunnally Miller, a noted philanthropist and dog breeder and trainer, gave the University of Georgia (UGA) CVM more than $7 million toward the construction of its Veterinary Medical Center [22]. John and Leslie Malone have recently provided Colorado State University (CSU) CVM with $42.5 million to help establish the C. Wayne McIlwraith Translational Medicine Institute. Horses owned by the Malones were successfully treated with orthopedic procedures developed by McIlwraith and other CSU faculty [23]. Construction of the research institute was further supported by a $20 million gift from Princess Abigail K. Kawananakoa of Hawaii, whose horses also received orthopedic care from McIlwraith. Other veterinary colleges report that their major donors also provide funds to endow individual faculty. This was the case with both the Malones’ and Stanton’s gifts to CSU and UGA, respectively. This experience illustrates how donors help direct gifts to different types of activities.

Philanthropic giving at U.S. medical schools has been the subject of limited research efforts. Engagement of medical school faculty has been assessed in several research studies [23,24,25,26]. Physician participation in fund raising efforts has been shown to increase the frequency and size of these donations [24]. Physician surveys have shown that many physicians at referral centers are aware of their institution’s fundraising activities. For example, a survey of medical oncologists at forty Comprehensive Cancer Centers showed that most (71%) had been exposed to their institution’s fundraising/development staff and nearly half (48%) were taught how to identify potential patient donors [26]. Most medical oncologists report that although they were aware of fundraising efforts only 26% had received information about ethical guidelines for soliciting donations from their patients [26]. Training is critical to successful fundraising since physicians generally lack formal training in how to respond to inquiries from grateful patients about philanthropy [24]. When compared with either email-based training or lectures, one-on-one interactions between development officers and physicians were most effective in training physicians about grateful patient philanthropy [24]. Some studies have shown gender and age differences in the rate at which physicians solicit gifts from their patients. For example, one study showed that increased years in practice and male gender were associated with an increased likelihood of soliciting donations from patients [26]. This study also showed that most (77%) responding oncologists were more likely to participate in fundraising for the disease or diseases they treat when compared with fundraising activities for their institution [26]. Excellent patient care has been identified by physicians as an important factor in grateful patient philanthropy [25]. We are not aware of similar studies being performed in veterinary medicine.

Another factor that needs to be considered relates to ethical fundraising. To our knowledge, this topic has not been addressed in the veterinary profession; however, it has been considered within medical schools. For example, a survey of physicians regarding attitudes towards patient philanthropy identified several potential ethical concerns including possible exploitation of vulnerable patients, providing differential care based on a patient’s level of giving, and concerns about patient confidentiality and trust [4,27,28]. It is currently unknown whether faculty at veterinary colleges share these concerns. Additional ethical issues have been raised among medical school faculty and administrators regarding philanthropy and the funding of research [29]. This concern arises since some research projects that are funded as a result of philanthropic giving are not peer reviewed as extensively as would occur through more traditional funding avenues. Based on experiences with medical schools, policies regarding philanthropic giving should be developed for veterinary schools and disseminated to all staff.

## 5. Conclusions and the Path Forward

Once established, development programs can significantly contribute to the financial health and stability of a veterinary college. For example, donor gifts to NC State CVM currently represent 23% of the total annual income revenue of the college in the fiscal year 2018-19; however, many of these gifts represent bequests, which will be realized in the future. Excluding documented bequest expectancies, new gifts and commitments account for approximately 7% of the college’s annual budget. In comparison, state appropriations, extramural research grants, and veterinary hospital income represent approximately 40%, 11%, and 26% of total revenue income to NC State CVM, respectively. New gifts and commitments to NC State CVM are higher than that reported by many U.S. AAMC-accredited medical schools [5]. Many gifts to NC State CVM are restricted/designated and support student scholarships, faculty, and equipment purchases. The prevalence of restricted giving follows a national trend that suggests donors want to earmark their gifts to a particular faculty member or program rather than a general purpose fund [19].

Grateful client philanthropy is an important part of academic veterinary medicine. As is the case in human medicine, this form of financial support is often restricted, transformative, and catalyzes innovation in areas of high institutional need. Grateful client programs at veterinary colleges should build on the collective experiences in human and veterinary medicine. It is critical that as veterinary colleges thoughtfully adopt best practices and policies to support development staff and faculty in their fundraising efforts that also allow them to maintain ethical and professional standards associated with the veterinarian-client-patient relationship. Development officers in veterinary colleges may need to develop workshops and other training opportunities in order to engage veterinary faculty in fundraising activities. There is also a critical need to develop empirical research related to grateful client philanthropy and its role in veterinary medicine.

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
