# Peer review of "Grateful Client Philanthropy and Veterinary Medicine: Experiences at North Carolina State University"

_vetsci, 2019, doi:10.3390/vetsci6020044_

Round 1
Reviewer 1 Report
Thank you for providing this opportunity to review and participate. Attached are suggested edits.

Author Response
Dear Reviewer 1,
Please accept our sincere thanks for your excellent review of our manuscript. We very much appreciated your thoughtful edits and comments throughout the document. As requested, we have provided a point-by-point response to each of the comments in the table format provided.
All the best, Dianne Dunning

Reviewer 2 Report
This is a very well-written and logically structured paper on a novel topic that should be of great interest to the journal's readers. It cites and discusses all the relevant literature on this topic, demonstrating mastery of that literature and bringing material from other disciplines (notably Philanthropic Studies) to the attention of a new audience.
I only have a few very minor suggestions that might improve the clarity for an international and multi-disciplinary readership:
p.2, lines 66 & 82 – might the word ‘article’, ‘paper’ or ‘Discussion paper’ be more suitable than ‘manuscript’ once this is published?
p.2, line 86 – designating non-major gifts as ‘Common gifts’ isn’t normal practice within the philanthropy/fundraising literature, nor does it seem accurate given this includes gifts up to $49,999. The most common alternative phrase that I’m aware of is ‘Core gifts’ but I’ve also seen people use ‘Mass gifts’, ‘Annual Gifts’ (though I appreciate there’s a 2-year payment option for Coats of Excellence) or the negative formulation of ‘Non-major Gifts’.
p.3, lines 97 & 98 – can you add an explanation of what a “paver” is?
p.5, line 222 – should ‘oncologist’ be plural?
p.6, lines 252-255 – these lines are a repetition of text from p.4, lines 187-191, can you re-phrase?
p.6, line 256 – I suggest moderating the word ‘essential’ (e.g. to ‘important’) as you note that many veterinary medicine schools, such as in Europe, do not achieve substantial philanthropic support.
p.6, line 260 – a comma after ‘…patient giving’ would make this sentence clearer. I had to re-read it a couple of times to catch the meaning!
Major Gift section - Clearly the R. B. Terry Charitable Foundation is central to NC State CVM’s success with grateful Client Philanthropy. Are you able to provide any more detail about the process of developing and growing the relationship with this donor, and with his descendants and foundation trustees since his death? This would also help to flesh out the ‘best practice that you cite and refer to in the ‘General Discussion’ section of your paper.
Author Response
Dear Reviewer #2,
Thank you for your comments and review of our manuscript. We very much appreciated the time and effort in reading and evaluation of the document. As requested, please find our point-by-point response to the suggestions.
Sincerely,
Dianne Dunning
